Genome-wide association study uncovers new genetic loci and candidate genes underlying seed chilling-germination in maize

Zhang Yinchao
Liu Peng
Wang Chen
Zhang Na
Zhu Yuxiao
Zou Chaoying
Yuan Guangsheng
Yang Cong
Gao Shibin
Pan Guangtang
Ma Langlang sxyljxml@163.com
Shen Yaou shenyaou@sicau.edu.cn
Maize Research Institute, Sichuan Agricultural University , Chengdu , Sichuan , China
Hill Camilla
Electronic publication date: 2021 Jun 28
Publication date: 2021
Volume: 9
Electronic Location ID: e11707
Received 2021 Mar 4; Accepted 2021 Jun 8
Copyright: ©2021 Zhang et al.
Copyright year: 2021
Copyright holder: Zhang et al.
License: This is an open access article distributed under the terms of the Creative Commons Attribution License, which permits unrestricted use, distribution, reproduction and adaptation in any medium and for any purpose provided that it is properly attributed. For attribution, the original author(s), title, publication source (PeerJ) and either DOI or URL of the article must be cited.
License URL: https://creativecommons.org/licenses/by/4.0/

Keywords: Maize, Chilling stress, Seed germination, Genome-wide association study, Candidate genes

Funding: National Natural Science Foundation of China 31871637 32072073 Sichuan Science, and Technology Program 2021JDTD0004 2021YJ0476 2019YJ0511 This work was supported by the National Natural Science Foundation of China (No. 31871637 and 32072073) and the, Sichuan Science, and Technology Program (No. 2021JDTD0004, No. 2021YJ0476 and No. 2019YJ0511). The funders had no role in study design, data collection and analysis, decision to publish, or preparation of the manuscript.

==============================
As one of the major crops, maize (Zea mays L.) is mainly distributed in tropical and temperate regions. However, with the changes of the environments, chilling stress has become a significantly abiotic stress affecting seed germination and thus the reproductive and biomass accumulation of maize. Herein, we investigated five seed germination-related phenotypes among 300 inbred lines under low-temperature condition (10 °C). By combining 43,943 single nucleotide polymorphisms (SNPs), a total of 15 significant (P < 2.03 ×  10-6) SNPs were identified to correlate with seed germination under cold stress based on the FarmCPU model in GWAS, among which three loci were repeatedly associated with multiple traits. Ten gene models were closely linked to these three variations, among which Zm00001d010454, Zm00001d010458, Zm00001d010459, and Zm00001d050021 were further verified by candidate gene association study and expression pattern analysis. Importantly, these candidate genes were previously reported to involve plant tolerance to chilling stress and other abiotic stress. Our findings contribute to the understanding of the genetic and molecular mechanisms underlying chilling germination in maize.

Introduction

Maize (Zea mays L.) is one of the major food, economic and energy crops globally, which is mainly distributed in tropical and temperate regions. Seed germination rate and seedling emergence rate are closely related to crop yield. With the environmental changes, chilling stress caused by the low temperatures (0−15 °C) has become one of the main abiotic stresses that affects seed germination, the reproductive and biomass accumulation of maize (Grieder et al., 2012). In high latitude or mountain regions, prolonging the growth period in early spring can increase maize productivity by accumulating more biomass (Grieder et al., 2012). However, seed germination is severely affected by cold spell in later spring, which then results in lower yield (Marocco, Lorenzoni & Fracheboud, 2005). Improving seed vigor and germination ratio are essential to ensure the successful emergence of seedlings and normal development for early-planting maize.

Chilling stress can disrupt normal metabolism, causing cell membrane hardening, the accumulation of ROS (reactive oxygen species), and protein instability (Guo, Liu & Chong, 2018). A number of studies have been focused on chilling stress in plants, which revealed how low temperature affects plant development and how plants resist freezing stress by sensing low temperature signals and initiating a series of response mechanisms. The CBF (C-repeat/DREB binding factors)—COR pathway is a well characterized cold-stress signaling pathway (Jia et al., 2016; Medina et al., 1999; Novillo, Medina & Salinas, 2007; Park et al., 2015; Zhao et al., 2016). In addition, a series of COR genes have also been identified, including ERD (early responsive to dehydration), RAB (responsive to abscisic acid), LTI (low temperature-induced) and KIN (cold-induced). The proteins encoded by these genes, and the osmolytes (glycine, proline, sucrose, and betaine) can stabilize cell membrane structure, remove excess ROS, and stabilize osmotic pressure, thereby improving plant tolerance to freezing stress (Cook et al., 2004; Gilmour et al., 1998; Guy et al., 2008). The MAP protein kinase family plays an important role in the response to chilling stress signals, through regulating its stability by phosphorylating ICE1 (Inducer of CBF Expression1). In Arabidopsis and rice, MAP kinase cascades, MPK3, MPK6 and OsMAPK3 were proven to modulate ICE1 stability and thereby regulate the response of plants to chilling stress (Li et al., 2017; Zhang et al., 2017; Zhao et al., 2017).

The genetic and molecular mechanisms on seed germination traits under chilling stress are still obscure, although a few potential cold-tolerant genes and QTL have been identified in maize (Huang et al., 2013; Hund et al., 2004; Leipner et al., 2008). In this study, we used a maize association panel composed of 300 inbred lines to explore the genetic architecture and candidate genes responsible for chilling tolerance of maize during seed germination. We aim at (1) evaluating the phenotypes of seed germination under chilling stress among these inbred lines; (2) identifying the significant SNPs associated with maize seed germination in chilling stress; (3) revealing the candidate genes and the intragenic variants affecting chilling tolerance of maize during seed germination.

Materials and Methods

Plant materials and germination experiment

To reflect the effects of low-temperature on seed germination, an association panel consisting of 300 inbred lines (Table S1) that were collected from the Southwest China breeding program (Zhang et al., 2016) were used to investigate the germination-related traits under chilling stress. The panel was planted in Chongzhou (CZ, E103° 67 ′, N30° 63 ′) of Sichuan Province (April 2018–August 2018) and the kernels were harvested at physiological maturity stage. The seeds with similar size and fullness from each line were sterilized with 10% (v/v) H2O2 for 30 min, washed with distilled water three times, incubated in saturated CaSO4⋅2H2O for 6 h, and finally washed by distilled water three times. A total of 30 seeds were germinated in a petri dish (130*130 mm) that was beforehand placed with a piece of sterilized wet filter paper on the bottom. Three replicates were conducted for each line. Following the ISTA protocol, seed germination experiment was conducted in a cultivation box (GUANDONG THK SCIENTIFIC INSTRUMENT CO., LTD) under 16 h/ 8 h, light/dark at 10 °C (Eagles & Hardacre, 1979; Silva-Neta et al., 2015) (Fig. S1).

Phenotyping and data analysis

In this study, the radicle emerging from the coat was defined as seed germination. A total of five germination-related traits were investigated including: (1) germination rate (the percentage of germinated seed number to total seed number) at 5 d (FG); (2) germination ratio at 10 d (TG); (3) root length at 10 d (RL); (4) shoot length at 10 d (SL) and (5) ratio of root length to shoot length (RRS).

For each trait, SPSS statistics version 20.0 software (https://www.ibm.com/support/pages/node/230551) was used to process the phenotypic data, including the range, mean, and standard deviation. SAS v 9.3 was used to estimate the broad heritability. The ‘PerformanceAnalytics’ package in R software was employed to analyze the phenotypic correlations. The analysis of variance (ANOVA) was performed in R software using the package ‘lme4’. The broad-sense heritability (HB2) estimations for each trait was calculated as following: HB2 = σ G2/ σ P 2 (Pace et al., 2015), σ G2 = ((MSG –MSE)/n) and σ P2 = ((MSG –MSE)/n + MES) are estimates of the genetic and phenotypic, respectively, where MSG and MSE represent the mean square for genotypes and the mean square error, respectively, n is the number of independent replications.

Genome-wide association study (GWAS)

The whole-genome SNPs of the 300 accessions were corresponded to the previous study (Zhang et al., 2016). The SNPs with missing rate (<20%) and minor allele frequency (MAF > 0.05) were retained by filtering the 56,110 SNPs, resulting in a final set of 43,943 high-quality SNPs. Independent tests (Meff_G) (Johnson et al., 2010) and multiple tests were conducted using simpleM in R studio (ver. 3.4.1) (Gao, Starmer & Martin, 2008) to calculate the effective number (N) of the SNPs in this study. The P-value (0.05/N) was set as a significance threshold. Herein, Meff_ G = 24,630, and thus, the P-valu e = 0.05∕24, 630 = 2.03 ×10−6. Based on the observation of P-values in different models under Q −Q diagrams (Kaler et al., 2017), the most suitable model FarmCPU was selected in this study. According to our previous study, the 300 lines were divided into three subpopulations (Zhang et al., 2020b), thus, “PCA = 3” was selected to perform GWAS in this study.

Candidate gene association study

According to our previous study, the LD of association panel was approximately 220 Kb (Liu et al., 2020). The genes between the 220Kb upstream and 220Kb downstream of the significant SNPs (P <  2.03 ×10−6) were therefore functionally annotated based on B73 AGPv4 genome (https://www.maizegdb.org/genome/assembly/Zm-B73-REFERENCE-GRAMENE-4.0). Among them, the gene models that were simultaneously associated with multiple chilling-germination traits were considered as the prioritized candidate genes in this study. To identify the intragenic variations associated with chilling-germination, sixty-three lines from the association pool were randomly selected for PCR-amplification of these candidate genes.. The primers of PCR amplification are shown in Table S2. Sequence alignment was conducted using DNAMAN software and DnaSP v5.0 was applied to identify sequence diversity (Zhang et al., 2020b). Using the GLM model of the GAPIT software (Lipka et al., 2012), the nucleotide variants with MAF ≥ 5% were retained for association analysis of these candidate genes, with P < 0.05 set as the significance level (Asekova et al., 2021; Li et al., 2011; Mamidi et al., 2014).

Analysis of superior alleles

Herein, an allele related to a higher value of these five traits (FG, TG, RL, SL, and RRS) was defined as a favorable allele. For each significant SNP, the ratio of elite allele was calculated by using the formula as follows: RE (%) = N (E)/ N (T) ×100%. Herein, RE is the ratio of elite allele, N (E) and N (T) represent the number of lines containing elite alleles and the total number of lines, respectively. We employed the ‘pheatmap’ package in the R software (Mellbye & Schuster, 2014) to display the distributions of superior alleles in each elite line.

RNA extraction and quantitative real-time PCR

According to the germination phenotypes and the elite haplotypes of four candidate genes, the lines SCL127 (AT/AC/AC/CG) and SCL326 (CC/CG/GG/TC) were selected for qRT-PCR (quantitative real-time PCR). In detail, maize seeds were germinated under 16 h/ 8 h, light/dark at 10 °C and the seed samples were collected at 0 h, 12 h, 24 h, 72 h, 120 h, with the normal condition (16 h/ 8 h, light/dark at 26 °C) as control. Total RNA was isolated from these samples using TRIzol (Novoprotein, Shanghai, China) according to the manufacturer’s instructions. For each sample, 1.5 µg of RNA was reversely transcribed to cDNA with NovoScript® Plus All-in-one 1st Strand cDNA Synthesis SuperMix (gDNA Purge) (Novoprotein, Shanghai, China). qRT-PCR was performed using an Applied Biosystems™ 7500 (Thermo Fisher, WM, USA) with NovoStart® SYBR qPCR SuperMix Plus (Novoprotein, Shanghai, China). To standardize the results, ZmActin1 (Zm00001d010159) was used to serve as a control (Wu et al., 2019) . Three biological replicate experiments were carried out for each sample. The primers that were used in qRT-PCR are summarized in (Table S3).

Results

Phenotypic descriptions for maize chilling-germination

The phenotypic performances were evaluated for five chilling-germination traits among the 300 inbred lines, namely FG, TG, RL, SL, and RRS (Table 1).The skewness values of these traits were between −1 and 1 (Table 1), indicating that these traits followed a normal distribution and conformed to the characteristics of quantitative traits. FG varied from 0%–60% with the mean of 17% and the SD (standard deviation) of 14%. SG ranged from 4%–96% with the mean of 31% and the SD of 22%. Varying from 0.50−4.67 cm, RL had an average value of 2.19 cm with the SD of 1.04 cm. SL had a range from 0.40 −1.73 cm with the SD of 0.35% and the mean of 0.91 cm (Table 1). Those phenotypic difference among the panel suggested that chilling stress exerted distinct effects on seed germination for the various lines. Significantly (P < 0.01) positive correlations were observed between each pair of FG, TG, RL, and SL, among which TG and FG showed the strongest correlation with the correlation coefficient of 0.81. However, RRS displayed a significantly (P < 0.01) negative correlation with each of the other traits except SL (Fig. 1).

Table 1 Phenotypic performance of seed chilling-germination among inbred lines.

Trait	Range	Mean	SD	CV	Kurtosis	Skewness	HB2	
FG (%)	0–60	0.17	0.14	0.77	0.30	0.97	88.00	
TG (%)	4–96	0.31	0.22	0.65	0.07	0.91	91.07	
RL (cm)	0.50–4.67	2.19	1.04	0.45	−0.49	0.64	96.53	
SL (cm)	0.40–1.73	0.91	0.35	0.37	−0.96	0.33	88.33	
RRS (%)	12–111	0.47	0.19	0.42	0.56	0.85	86.96	
Notes.

FG Germination rate at 5 d

TG Germination rate at 10 d

RL Root length at 10 d

SL Shoot length at 10 d

RRS Ratio of root length to shoot length

SD Standard deviation

CV Coefficient of variation

Figure 1 Distributions of phenotypic frequency and correlations between five germination traits.

Phenotypic frequency distributions for each trait are illustrated as histograms in the center diagonal. Scatter plots of correlations and the numerical correlation coefficients between each two traits are shown in the areas below and above the diagonal, respectively. The red lines in the scatter plots represent the correlation trends. FG, Germination rate at 5 d; TG, Germination rate at 10 d; RL, Root length at 10; SL, Shoot length at 10 d; RRS, Ratio of root length to shoot length.

Significant SNPs and candidate genes for germination under chilling stress

In this study, we employed the FarmCPU model to detect the associations between the SNPs and the germination traits under chilling stress (Fig. S2). A total 15 significant (P < 2. 03 × 10−6) SNPs were identified for these investigated traits (Table 2 and Fig. 2). Among them, PZE-104042136 was associated with both RL and SL, and PZE-107018981 was identified to correlate with FG and TG. Intriguingly, PZE-108063385 was associated with all the five traits. Based on the LD decay (220 Kb) of the association panel, a total of 138 gene models were identified for these significant SNPs (Table S4). For the three SNPs associated with multiple traits, 10 gene models were found within the LD regions including several genes involving seed dormancy and germination, plant development and cold stress pathways (Table 3). For PZE-104042136, only one candidate gene (Zm00001d050021) was detected, which encodes an abscisic acid 8′-hydroxylase3 (abh3). A total of 4 candidate genes (Zm00001d019116, Zm00001d019123, Zm00001d019117 and Zm00001d019122) were identified for PZE-107018981, among which Zm00001d019116 and Zm00001d019123 were annotated as ethylene-responsive transcription factor RAP2-11, and survival protein SurE-like phosphatase/nucleotidase, respectively, whereas Zm00001d019117 and Zm00001d019122 both encode unknown proteins. PZE-108063385 was associated with 5 gene models, of which Zm00001d010454, Zm00001d010458, and Zm00001d010459 were individually annotated as mannosyl-oligosaccharide 12-alpha-mannosidase MNS3, protein kinase superfamily protein, and putative CBL-interacting protein kinase family protein, respectively. Zm00001d010455 and Zm00001d010456 were not functionally annotated based on the current B73 genome information.

Table 2 Chilling germination-associated SNPs identified in this study.

SNP	Chr.	Physical position	Allele frequency	Trait	P.value	
PZE-108086767	1	220,338,489	46.21% (A), 32.85% (C)	RRS	1.59E−06	
PZE-102118136	2	158,379,376	85.56% (A), 13.36% (G)	RRS	1.01E−06	
PZE-102193367	2	236,235,981	40.07% (A), 58.48% (G)	SL	9.85E−07	
PZE-103096466	3	156,304,805	16.25% (A), 75.81% (C)	FG	2.71E−07	
SYN8915	3	196,150,129	15.52% (A), 75.45% (G)	TG	9.73E−07	
PZE-104042136	4	58,031,648	41.52% (A), 53.79% (C)	RL, SL	3.67E−08, 1.03E−06	
PZE-105052784	5	48,384,424	36.10% (A), 58.12% (C)	FG	1.82E−07	
PZE-105056721	5	54,915,313	43.32% (A), 53.79% (C)	TG	5.95E−07	
PZE-107018981	7	17,240,858	74.01% (A), 16.97% (G)	FG, TG	8.10E−08, 1.67E−07	
SYN19853	7	161,949,846	24.19% (A), 74.73% (C)	RL	1.17E−06	
SYN13495	7	125,974,879	57.40% (A), 37.91% (C)	FG	3.86E−08	
PZE-108063385	8	113,291,192	54.15% (A), 44.40% (G)	FG, TG, SL, RL, RRS	3.86E−08, 8.10E−08, 6.31E−07, 3.67E−08, 4.78E−07	
PZE-108134421	8	173,977,740	31.41% (A), 57.40% (G)	TG	4.27E−07	
SYN2994	8	112,768,552	44.40% (A), 52.35% (G)	RL	4.18E−07	
PZB02554.2	10	142,190,907	31.33% (A),66.00% (C)	TG	1.10E−06	
Notes.

Chr., chromosome.

Figure 2 Manhattan plots of five chilling-germination traits using FarmCPU model.

FG, Germination rate at 5 d; TG, Germination rate at 10 d; RL, Root length at 10 d; SL, Shoot length at 10 d; RRS, Ratio of root length to shoot length. The yellow lines show the significant P-threshold of 2.03 × 10−6; the blue dots represent the significant SNPs; the color scales denote marker density.

Table 3 Genes located in the LD regions of the multiple traits-associated SNPs in this study.

SNP	Candidate gene	Chr.	Trait	Position (bp)	Functional annotations	
PZE-104042136	Zm00001d050021	4	RL, SL	57,945,650-57,951,076	Abh3; abscisic acid 8′-hydroxylase3	
PZE-107018981	Zm00001d019116	7	FG, TG	17,143,991-17,145,013	Ethylene-responsive transcription factor rap2-11	
Zm00001d019117	7	17,179,984-17,182,116	Unknown	
Zm00001d019122	7	17,254,408-17,255,528	Unknown	
Zm00001d019123	7	17,308,289-17,314,146	Survival protein sure-like phosphatase/nucleotidase	
PZE-108063385	Zm00001d010454	8	FG, TG, SL, RL, RRS	113,176,478-113,180,365	Mannosyl-oligosaccharide 12-alpha-mannosidase mns3	
Zm00001d010455	8	113,205,914-113,208,800	Unknown	
Zm00001d010456	8	113,347,031-113,351,017	Unknown	
Zm00001d010458	8	113,414,362-113,438,862	Protein kinase superfamily protein	
Zm00001d010459	8	113,500,500-113,503,393	Putative cbl-interacting protein kinase family protein	
Notes.

Chr. chromosome

FG Germination rate at 5 d

TG Germination rate at 10 d

RL Root length at 10 d

SL Shoot length at 10 d

RRS Ratio of root length to shoot length

Utilization of favorable alleles in maize elite lines

Thirty inbred lines that have been used as the parents of commercial varieties were included in the association panel, which enables us to evaluate the utilization of the superior alleles in maize breeding programs (Fig. 3). In the present study, an allele correlating with higher SL, RL, FG, TG or RRS were defined as a favorable allele for a given SNP. For the 15 significant SNPs identified in our study, we separately calculated the favorable ratio of each SNP in 30 lines, which ranged from 10% (PZE-103096466) to 96.67% (PZE-102118136). Five loci (SYN2994, PZE-104042136, SYN13495, PZE-107018981 and PZE-102118136) have an excellent allele frequency of ≥50%. However, the frequency of superior allele in five loci (PZE-103096466, SYN8915, PZE-108134421, PZE-108063385, PZB02554.2) was ≤ 20%. The frequency of favorable alleles in 30 elite lines ranged from 26.67% to 60%, and only four (13.3%) of the 30 lines contained 60% favorable alleles. The lines containing 7–9 superior alleles showed the higher average phenotypic values, with 17.44% in FG, 31.78% in TG, 1.83 cm in RL, 0.85 cm in SL, and 50.31% in RSR, whereas the lines with 4–6 superior alleles had the lower averages of 13.70%, 24.89%, 1.81 cm, 0.81 cm, and 47.85% in FG, TG, RL, SL, and RSR, respectively. The results suggested that these superior alleles had additive effects on the chilling-germination traits. In the future, chilling germination of commercial varieties could therefore be improved by integrating more favorable alleles into the elite maize lines.

Figure 3 The distributions of the favorable alleles in the 30 elite lines of maize.

Dark and white colors represent superior and inferior alleles, respectively.

Intragenic variations affecting germination under chilling stress

For the 6 functionally annotated candidate genes, we further identified the intragenic variations that influenced seed germination under chilling stress by candidate gene association analysis. A total of 70 variations (67 SNPs and 3 InDels) were found within the gene bodies and their promoter regions (the upstream 2000bp) of the 6 genes (Table S5). Eight SNPs were detected to significantly (P < 0.05) correlate with chilling-germination, involving two (SNP-4-60282052 and SNP-4-60288956) in Zm00001d050021, two (SNP-8-115913550 and SNP-8-115914606) in Zm00001d010454, two (SNP-8-116147619 and SNP-8-116148568) in Zm00001d010458, one (SNP-7-17937736) in Zm00001d019123 and one (SNP-8-116262789) in Zm00001d010459.

The two variations SNP-8-116147619 and SNP-8-116148568 resided in the promoter and the intron of Zm00001d010458, respectively, among which SNP-8-116147619 was associated with SL and RL and SNP-8-116148568 was related to all the five traits. Two haplotypes (Hap1: AC, Hap2: GC) formed in these 63 lines according to the two SNPs, and all the phenotypic values of Hap1 were significantly lower than those of Hap2 (Fig. 4). For Zm00001d050021, SNP-4-60288956 that was located in the 3′-UTR was associated with SL and RL, whereas SNP-4-60282052 that resided in the promoter was found to relate to SL. Based on the two significant SNPs, the 63 lines were classed into three groups, with one group containing the alleles CG (Hap1) and the other groups including the alleles CC (Hap2) and TC (Hap3), respectively. T-test indicated that these lines with Hap3 had significantly (P < 0.05) larger SL and RL than those with Hap1 and Hap2 (Fig. S3). For Zm00001d010454, SNP-8-115913550 and SNP-8-115914606 were both located in the promoter and were associated with SL. According to the genotypes of the two variations, these 63 lines were divided into three groups, namely Hap1 (CC), Hap2 (AC). Hap2 showed a significantly larger SL value than Hap3 (Fig. S4). SNP-8-116262789 was located in the promoter of Zm00001d010459, which was significantly associated with SL. Referring to the genotypes of SNP-8-116262789, these lines were clustered into two groups, with group I containing the allele C and group II having A. Analysis of variance showed that SL was significantly (P < 0.05) smaller in group I than in group II (Fig. S5). However, no significantly phenotypic difference was observed between diverse haplotypes of Zm00001d019123 among these lines. Consequently, Zm00001d010458, Zm00001d050021, Zm00001d010454, and Zm00001d010459 were prioritized to be the causal genes for chilling-germination in this study.

Figure 4 Significant variants located within the promoter and the intron of Zm00001d010458.

(A) Dark dots above the dotted line display the significantly ( P < 0.05) associated SNPs. The structure of the gene Zm00001d010458 is displayed in the middle. The extrons are represented by filled dark boxes, the promoter and 5′-UTR are shown by the left dark line, the introns were denoted by the dark lines between the filled dark boxes, and the 3′-UTR are displayed by the right dark line. The bottom image shows the pairwise LDs between the markers. (B) Comparison of the five phenotypes between two haplotypes. **, P < 0.01 level; ***, P < 0.001. Hap, haplotypes; FG, Germination rate at 5 d; TG, Germination rate at 10 d; RL, Root length at 10 d; SL, Shoot length at 10 d; RRS, Ratio of root length to shoot length.

Candidate gene expressions during chilling-germination

Significant variations in the promoters of these four candidate genes were associated correlate with these chilling-germination traits, which were assumed to influence expression levels of these genes between chilling tolerant lines and sensitive lines. We therefore performed qRT-PCR to verify the expression difference of these genes above between SCL127 (a chilling-senstive line) and SCL326 (a chilling-tolerant line) during chilling-germination (0 h, 12 h, 24 h, 72 h, and 120 h). Under control condition, the expression of Zm00001d010459 was up-regulated from 0 h to 24 h and then down-regulated during the following stages in both SCL326 and SCL127. However, the expression was higher in SCL326 than in SCL127 at all the stages except 0 h in CK. In chilling stress, the expression of Zm00001d010459 was continuously increased throughout the whole cold treatment in both the lines, and the expression abundance was consecutively higher in SCL326 than in SCL127 at all treatment stages (Fig. 5A). Under CK condition, the expression of Zm00001d010454 was not significantly regulated during the developmental stages from 0 h to 120 h, and no significant difference of expression levels was observed between the two lines. Notably, in both lines, Zm00001d010454 was slightly down-regulated from 0 h to 12 h, then sharply up-regulated from 12 h to 72 h and subsequently returned to a lower level at 120 h under chilling stress. Moreover, the expression abundance of Zm00001d010454 was significantly higher in SCL326 than in SCL127 from 24 h to 120 h under chilling condition (Fig. 5B). In the chilling-sensitive line SCL127, the expression of Zm00001d010458 was only slightly regulated by the stress. However, in the tolerant line SCL326, its expression was abruptly enhanced by more than 20 folds with the low-temperature treatment from 12 h to 100 h (Fig. 5C). Interestingly, compared to CK, Zm00001d050021 was down-regulated under the chilling stress in both the lines, whereas the decreased range of the expression level was larger in SCL326 than in SCL127 (Fig. 5D). These observations in the expression patterns between the contrasting lines supported the associations between variations in the promoters and the chilling-germination traits.

Figure 5 Expression patterns of four candidate genes in the chilling-sensitive line (SCL127) and tolerant line (SCL326) under cold stress (A-D).

The RNA from germinating seeds of SCL127 and SCL326 were isolated for qRT-PCR to obtain the expression patterns of candidate genes in the contrasting lines under low-temperature treatment (0 h, 12 h, 24 h, 72 h, and 120 h), with the normal condition as control check. Three biological replicate experiments were carried out for each sample. Error bars indicate SD values.

Discussion

Using the association panel to identify QTL for seed germination traits under chilling stress

High heritability and abundant phenotypic variations were both necessary to detect the genetic architecture of target traits (Liu et al., 2020; Zhang et al., 2020b; Zhang et al., 2020c). According to the population structure, the association panel was classified into tropical group, stiff stalk (SS) group and non-stiff stalk (NSS) group. Moreover, the association panel was demonstrated to have remarkable phenotypic variations in germination traits under low temperature with the higher coefficients of variations (CVs) (0.77, 0.65, 0.45, 0.37, and 0.42 for FG, TG, RL, SL and RRS, respectively) (Table 1). The observed CVs in this study were generally higher than those of the other traits among this panel in the previous studies (Li et al., 2020; Liu et al., 2020; Ma et al., 2018). Estimations of broad heritability using SAS v 9.3 indicated that the HB2 of these investigated traits ranged from 86.96%-96.56% (Table 1), suggesting that these chilling-germination traits were mainly genetically controlled. By GWAS, a total of 15 loci were significantly associated with the germination-related traits under low temperature. Within the LD region of these significant loci, some known genes involving the response of chilling or abiotic stress were identified including ABH4, cold-regulated plasma membrane protein, CBL-interacting protein kinase, and inducer of CBF expression 2 (Table S4). Collectively, these findings supported the high efficiency and reliability to use the accessions for dissecting the genetic basis of seed chilling-germination in maize.

Genetic architecture of low-temperature tolerance in maize

The previous study identified the QTL controlling the cold-related traits in maize (Table S6), uncovering 60 genetic loci responsible for the root, shoot and seed traits under chilling stress (Hund et al., 2004). QTL mapping using a maize F2:3 family identified 25 QTL for plant height, flowering, and biomass traits under cold stress across two years (Leipner et al., 2008). In an association panel consisting of 125 inbred lines, 43 SNPs were associated with 10 germination traits and seedling growth traits under chilling stress and control condition by GWAS (Huang et al., 2013). Hu et al. detected the genetic elements of chilling tolerance during maize seed germination using GWAS, revealing 17 genetic loci and 18 candidate genes associated with cold tolerance (Hu et al., 2017). In a panel of 222 maize inbred lines, 30 SNPs were associated with low-temperature tolerance during maize seed germination (Zhang et al., 2020a). To better understand the genetic architecture of chilling tolerance in maize, we compared the physical positions of the genetic loci between the present study and the previous studies. As shown in Table 2, five SNPs (PZE-102118136, PZE-102193367, PZE-105052784, PZE-105056721 and PZE-107018981) that were associated with RRS, SL, TG, SG, and SG, respectively, by this study situated within the QTL for maize chilling-related seedling reported previously (Leipner et al., 2008). An overlap of LD regions existed between the significant SNP PZE-108063385 (Chr8:113,291,192) in our study and the cold germination-associated SNP PZE-108064544 (Chr8:113,952,611) reported by Zhang et al. (2020a); Zhang et al. (2020a). Four associated SNPs (SYN19853, PZE-103096466, PZE-104042136, and PZE-107018981) in this study resided in the previously reported QTL controlling root and shoot traits of maize seedlings under cold stress. Moreover, we detected two significant SNPs associated with RRS and SL, respectively, which were individually located in the LD of the two SNPs (SNP11 and SNP16) controlling root weight and root length (Huang et al., 2013). Combined these findings demonstrated the genetic overlaps of maize between different developmental traits under chilling stress.

Candidate genes involved in maize chilling-germination

A total of 138 gene models were identified in the LD regions of the 15 significant SNPs in this study, among which 10 genes (including 6 functionally annotated genes) were repeatedly associated with multiple chilling-germination traits and were thereby considered as the prioritized candidate genes mediating chilling germination in maize. More intriguingly, the intragenic variations from 4 (66.67%) candidate genes were further proven to influence these chilling-germination traits by candidate gene association studies and expression pattern analysis. Among them, Zm00001d010454 encodes a mannosyl-oligosaccharide 1, 2-alpha-mannosidase MNS3. The α-mannosidases are involved in abiotic plant stress tolerance by participating in protein folding, trafficking, and endoplasmic reticulum-correlating degradation in eukaryotic cells (Wang et al., 2020), which have been proven to regulate plant tolerance to salt stress (Liu et al., 2018; Von Schaewen, Frank & Koiwa, 2008; Wang et al., 2020). Zm00001d010458 was annotated as protein kinase superfamily protein, the protein kinases were extensively reported to confer plant tolerance to abiotic stresses as salinity, drought, and cold stress (Deng et al., 2013; Komatsu et al., 2007; Kong et al., 2011; Saijo et al., 2000). The encoded protein by Zm00001d010459 is a member of CBL-interacting protein kinase (CIPK) family, its homologous gene OsCIPK31 is involved in germination and seedling growth under abiotic stress conditions in rice(Piao et al., 2010). In wheat, CIPK23 positively participates in the response to ABA response and regulates drought stress tolerance (Cui et al., 2018). Similarly, ZmCIPK8 regulates maize response to drought and other abiotic stress via coordinating stress-related genes (Tai et al., 2016). Zm00001d050021 encodes an abscisic acid 8′-hydroxylase3, the enzyme plays a key role in the oxidative catabolism of ABA in Arabidopsis (Saito et al., 2004). It was previously reported that ABA regulated freezing tolerance and thereby cold acclimation in barley (Bravo et al., 1998). In wheat, ABA improving chilling tolerance through enhancing ROS scavenging system (Yu et al., 2020). Taken together, these findings suggested that these candidate genes were involved in maize chilling-germination.

Supplemental Information

Supplemental Information 1 Passport information of 300 inbred lines in this study

Click here for additional data file.

Supplemental Information 2 Primers used for PCR amplification of candidate genes

Click here for additional data file.

Supplemental Information 3 Primers used for qRT-PCR amplification of candidate genes

Click here for additional data file.

Supplemental Information 4 List of gene models within LD regions of the significant SNPs identified by GWAS

Click here for additional data file.

Supplemental Information 5 Results of candidate gene association analysis in this study

Click here for additional data file.

Supplemental Information 6 Our identified significant SNPs overlapping with QTLs/SNPs under cold-germination identified in previous studies

Click here for additional data file.

Supplemental Information 7 The germination of different types of maize lines under CK and chilling stress

Click here for additional data file.

Supplemental Information 8 Q–Q plot resulting from FarmCPU model for chilling-germination traits

Click here for additional data file.

Supplemental Information 9 Significant variants located within the promoter and the 3′-UTR of Zm00001d050021

(A) Dark dots above the dotted line display the significantly ( P < 0.05) associated SNPs. The structure of the gene Zm00001d050021 is displayed in the middle. The extrons are represented by filled dark boxes, the promoter and 5′-UTR are shown by the left dark line, the introns were denoted by the dark lines between the filled dark boxes, and the 3′-UTR are displayed by the right dark line. The bottom image shows the pairwise LDs between the markers. (B) Comparison of chilling-germination phenotypic performance between three haplotypes. * and *** represent the significant levels of P < 0.05 and P < 0.001, respectively. Hap, haplotypes; FG, Germination rate at 5 d; TG, Germination rate at 10 d; RL, Root length at 10 d; SL, Shoot length at 10 d; RRS, Ratio of root length to shoot length.

Click here for additional data file.

Supplemental Information 10 Significant variants located within the promoter of Zm00001d010454

(A) Dark dots above the dotted line display the significantly ( P < 0.05) associated SNPs. The structure of the gene Zm00001d010454 is displayed in the middle. The extrons are represented by filled dark boxes, the promoter and 5′-UTR are shown by the left dark line, the introns were denoted by the dark lines between the filled dark boxes, and the 3′-UTR are displayed by the right dark line. The bottom image shows the pairwise LDs between the markers. (B) Comparison of chilling-germination phenotypic performance between three haplotypes. ** represent the significant level of P < 0.05. Hap, haplotypes; FG, Germination rate at 5 d; TG, Germination rate at 10 d; RL, Root length at 10 d; SL, Shoot length at 10 d; RRS, Ratio of root length to shoot length.

Click here for additional data file.

Supplemental Information 11 Significant variant located within the promoter of Zm00001d010459

(A) Dark dot above the dotted line shows significant ( P < 0.05) SNP. The structure of the gene Zm00001d010459 is displayed in the middle. The extrons are represented by filled dark boxes, the promoter and 5′-UTR are shown by the left dark line, the introns were denoted by the dark lines between the filled dark boxes, and the 3′-UTR are displayed by the right dark line. The bottom image shows the pairwise LDs between the markers. (B) Comparison of chilling-germination phenotypic performance between two haplotypes. *** represent the significant level of P < 0.001. Hap, haplotypes; FG, Germination rate at 5 d; TG, Germination rate at 10 d; RL, Root length at 10 d; SL, Shoot length at 10 d; RRS, Ratio of root length to shoot length.

Click here for additional data file.

Supplemental Information 12 Genotype data

Click here for additional data file.

Supplemental Information 13 Phenotype

Click here for additional data file.

Supplemental Information 14 Raw data for Fig. 1

Click here for additional data file.

Supplemental Information 15 Raw data for Fig. 4

Click here for additional data file.

Additional Information and Declarations

Competing Interests

Author Contributions

Data Availability

The authors declare there are no competing interests.

Yinchao Zhang and Peng Liu performed the experiments, analyzed the data, prepared figures and/or tables, and approved the final draft.

Chen Wang, Na Zhang, Yuxiao Zhu, Chaoying Zou, Guangsheng Yuan, Cong Yang and Guangtang Pan performed the experiments, prepared figures and/or tables, and approved the final draft.

Langlang Ma and Yaou Shen conceived and designed the experiments, authored or reviewed drafts of the paper, and approved the final draft.

The following information was supplied regarding data availability:

The raw measurements are available in the Supplemental Files.

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
