# Peer review of "Genome-wide association study uncovers new genetic loci and candidate genes underlying seed chilling-germination in maize"

_PeerJ, doi:10.7717/peerj.11707_

## Round 0.1 · original submission · Minor Revisions

In addition to the reviewer's comments, please address the following points in your new version of the manuscript:

P3, L35: “Chilling stress can disrupt normal material metabolism” – remove “material”
P3, L36: “which systematically revealed how cold stress affects plant development”– remove “systematically”
P3, L38: “COR pathway was the best characterized among” – change to “COR pathway is a well characterized cold-stress signaling pathway”
P3, L48-49: References missing (particularly QTL studies showing associations with cold tolerance)
P4, material and methods: Passport information on 300 inbred lines required as supplemental table
P4, L81: change “Missing rate (< 20%) and minor allele frequency (MAF > 0.05) were retained” to “SNPs with missing rate (< 20%) and minor allele frequency (MAF > 0.05) were retained”
Page 5, L84: “The P-value (0.05/N) was set as a significance threshold.” – explain what N means
P5, L90-92: “the genes which located in the LD regions (220 Kb) of the significant SNPs (P < 2.03×10-6) were” – unclear how LD was calculated
P5, L92-93: “Among them, the gene models associated with multiple chilling-germination traits were considered as the prioritized candidate genes in this study” – unclear what “multiple chilling-germination traits” mean. How was this determined?
P5, L98: reference missing for GAPIT software
P5, L100: “Herein, an allele related to a higher value of these five traits was defined as a favorable allele.” – Which five traits? Mention explicitly
P5, L101: “For each significant SNP, the ratio of elite allele was denoted by the number of lines with elite alleles divided by the total number of lines” – For easy of understanding, write this up as a formula and define clearly
P6, L119: “including FG, TG, RL, SL, and RRS” – change “including” to “namely”, and introduce all abbreviations
P8, L162: “and all the phenotypic values of Hap1 were lower than those of Hap2 (Fig. 3).” – significantly lower?
P9 Discussion section: The Discussion section needs to be re-worker as large parts of the Discussion section repeat the results. The Discussion should instead provide an analysis of the results and clear, scientifically sound conclusions that are firmly based on the data analysis. The Discussion should relate clearly to the main hypothesis/question posed in the Introduction and should explain how the presented evidence substantiates the main claims of the paper.

·

Basic reporting

no comment

Experimental design

no comment

Validity of the findings

no comment

Additional comments

In this work,the authors used 43,943 SNPs and FarmCPU method to perform GWAS of five germination-related traits during seed chilling-generation in maize. They detected 15 significant signals under a Bonferroni multiple-test threshold, among which, they further analyzed 3 loci and 10 candidate genes associated with multiple traits. Overall, this manuscript was well organized, and I favor its acceptance after a few of minor editions.
1. The manuscript would be much improved after professional-English editing. The results section was particularly verbose.
2.Please clearly define the differences among chilling, cold and freezing stress and confirm the correct use of these three words in MS.
3.What were the criteria for selecting seeds (eg. size, weight...)? Also, germination experiments required a normal temperature control. It would be more convincing if some germinating pictures of different types of maize lines were provided.
4.Please provide the evidence that the chilling condition was chosen at 10℃.
5.Most figures weren’t at sufficient resolution as the request of Peer J.
6.Lines 136-137: I can’t find the LD decay regions with 220K in Conneely & Boehnke (2010). It should be described in Method or cited the relevant published article.
7.Lines 130-133: FarmCPU is a widely used GWAS method with high power and low false positive and false negative. That is well known and not an important point in MS. I suggest to delete this paragraph.
8.Fig. 3: The threshold was nearly 1.3, which was different from GWAS threshold (P<2.03*10-6). How to get the threshold?
9.Lines 207-208: “Statistical Analysis System v 9.3” should be abbreviated to “SAS v 9.3”. Alternatively, remove it here and describe it in Method instead.
10.Line 211: The name of the candidate genes “abh4” should be “ABH4”.
11.Section 4.2: This part compared the overlaps between present genetic loci of chilling tolerance and previous published data. Readers would be put off by searching for so much genetic information in so many articles. So I suggest making a list of previous studies.
12.Fig. 5: Heat maps shouldn't just have two colors. This title needs to be reconsidered.

Reviewer 2 ·

Basic reporting

I think that there are no problems with the authors' writing and format in the manuscript.

Experimental design

In this manuscript, the authors investigated five seed germination-related phenotypes such as the germination rate or growth under the cold stress among 300 inbred lines. By using these data, they performed the GWAS analysis and identified several SNPs that were associated with the cold-tolerant phenotypes. These SNPs gave positive effects to the expression of four genes in response to the cold stress. Their experimental design was clear.

Validity of the findings

I agreed that the ten genes that were associated with multiple traits were good candidates of the important genes for germination under the cold stress.

Additional comments

I listed several concerns below.

・The authors should transfer the description about Fig.5 to the Result section. It shows how important the SNPs that they identified are for germination under the cold stress. I think they should show how different the germination phenotypes under the cold stress are between the five elite lines and the others.

・In L170, the authors wrote that Hap2 and Hap3 showed significantly larger SL values than that of Hap1. Is it correct? I look like that there is rather a significant difference between Hap2 and Hap3 in their figure.

・It is difficult to see where three groups are divided in Table 2.

・I think that some “Figs.” are typo of “Fig. S”.

---

## Round 0.2 · Minor Revisions

Dear Authors,

please address the following comment:

A GWAS on cold-tolerance and germination in maize was published in 2020 but not discussed. https://pubmed.ncbi.nlm.nih.gov/32664856/.

The authors need to discuss their results in relation to the 2020 study.

Regards

---

## Round 0.3 · accepted · Accept

Dear authors,

As you have addressed the outstanding comment satisfactorily, your manuscript has been accepted for publication.